# A MYC-Driven Plasma Polyamine Signature for Early Detection of Ovarian Cancer

**DOI:** 10.3390/cancers13040913

**Published:** 2021-02-22

**Authors:** Johannes F. Fahrmann, Ehsan Irajizad, Makoto Kobayashi, Jody Vykoukal, Jennifer B. Dennison, Eunice Murage, Ranran Wu, James P. Long, Kim-Anh Do, Joseph Celestino, Karen H. Lu, Zhen Lu, Robert C. Bast, Samir Hanash

**Affiliations:** 1Department of Clinical Cancer Prevention, The University of Texas M. D. Anderson Cancer Center, Houston, TX 77030, USA; jffahrmann@mdanderson.org (J.F.F.); dm11018s@st.kitasato-u.ac.jp (M.K.); jvykouka@mdanderson.org (J.V.); jbdennis@mdanderson.org (J.B.D.); ENMurage@mdanderson.org (E.M.); RWu2@mdanderson.org (R.W.); 2Department of Biostatistics, The University of Texas M. D. Anderson Cancer Center, Houston, TX 77030, USA; EIrajizad@mdanderson.org (E.I.); JPLong@mdanderson.org (J.P.L.); kimdo@mdanderson.org (K.-A.D.); 3Department of Gynecological Oncology and Reproductive Medicine, The University of Texas M. D. Anderson Cancer Center, Houston, TX 77030, USA; jcelesti@mdanderson.org (J.C.); khlu@mdanderson.org (K.H.L.); zlu@mdanderson.org (Z.L.); 4Department of Experimental Therapeutics, The University of Texas M. D. Anderson Cancer Center, Houston, TX 77030, USA; rbast@mdanderson.org

**Keywords:** blood-based biomarkers, polyamines, ovarian cancer, early detection

## Abstract

**Simple Summary:**

There is a need for additional marker(s) to detect early-stage ovarian cancer that would augment the performance of CA125. Herein, we report a polyamine signature that is detected in the blood and that has value for detecting ovarian cancers at an early stage. The polyamine signature was able to complement CA125 in identifying more ovarian cancer cases that would have been missed by CA125 alone. Our validation of a polyamine signature provides compelling evidence for the value of blood polyamine metabolites as markers for ovarian cancer detection.

**Abstract:**

MYC is an oncogenic driver in the pathogenesis of ovarian cancer. We previously demonstrated that MYC regulates polyamine metabolism in triple-negative breast cancer (TNBC) and that a plasma polyamine signature is associated with TNBC development and progression. We hypothesized that a similar plasma polyamine signature may associate with ovarian cancer (OvCa) development. Using mass spectrometry, four polyamines were quantified in plasma from 116 OvCa cases and 143 controls (71 healthy controls + 72 subjects with benign pelvic masses) (Test Set). Findings were validated in an independent plasma set from 61 early-stage OvCa cases and 71 healthy controls (Validation Set). Complementarity of polyamines with CA125 was also evaluated. Receiver operating characteristic area under the curve (AUC) of individual polyamines for distinguishing cases from healthy controls ranged from 0.74–0.88. A polyamine signature consisting of diacetylspermine + N-(3-acetamidopropyl)pyrrolidin-2-one in combination with CA125 developed in the Test Set yielded improvement in sensitivity at >99% specificity relative to CA125 alone (73.7% vs 62.2%; McNemar exact test 2-sided P: 0.019) in the validation set and captured 30.4% of cases that were missed with CA125 alone. Our findings reveal a MYC-driven plasma polyamine signature associated with OvCa that complemented CA125 in detecting early-stage ovarian cancer.

## 1. Introduction

Currently, over 70% of patients with ovarian cancer present with advanced-stage (III-IV) disease, with dismal 5-year survival rates of less than 30%. Survival rates up to 70–90% can, however, be achieved with conventional surgery and chemotherapy, when disease is localized to the ovary (stage I) or pelvis (Stage II). [1,2] To-date, CA125 is the most investigated ovarian cancer early detection marker. [3] Neither CA125 nor transvaginal sonography (TVS) alone has adequate sensitivity or specificity for early detection. A two-stage strategy whereby rising CA125 prompts TVS in a limited fraction of women screened can achieve adequate specificity, but the sensitivity of CA125 is limited. [4] Moreover, only 80% of epithelial ovarian cancers express significant levels of CA125. [5] Thus, there is a need for an additional marker(s) to detect early-stage disease that would complement the performance of CA125.

Ovarian cancer and triple-negative breast cancer (TNBC) share common genomic features including MYC copy-number amplification. [6] MYC is an oncogenic driver in the pathogenesis of ovarian cancer. [7,8,9] We have previously demonstrated that MYC regulates the transcription of several polyamine metabolizing enzymes (PMEs) in triple negative breast cancer and that a plasma polyamine signature is associated with TNBC development and progression. [10] Herein, we tested whether a plasma polyamine signature would similarly be associated with ovarian cancer. We also evaluated whether polyamines, in combination with CA125, would improve classification performance compared to CA125 alone. 

## 2. Materials and Methods

### 2.1. Blood Samples

EDTA-plasma samples were obtained from the Normal Risk Ovarian Cancer Screening Study (NROSS) and stored at the MD Anderson Gynecologic Cancer Bank. All biospecimen were processed at a central site using a standardized protocol; EDTA-plasmas were stored in −80 °C until use. Ethical approval was obtained for these samples from the appropriate institutional review boards/ethic committees at MD Anderson and collaborating institutions under IRB protocol LAB04-0687. All participants had consent for the use of samples in ethically approved secondary studies.

Randomly selected healthy control plasmas were obtained from the NROSS cohort. Over the last two decades, the NROSS trial has involved 6379 postmenopausal women over the age of 50 at average risk for developing ovarian cancer and who have been followed with annual CA125 measurements and who have been referred for transvaginal ultrasound and gynecological evaluation if CA125 values increase from each individual’s baseline as judged by the Risk of Ovarian Cancer Algorithm (ROCA). Healthy controls were at least 12 months postmenopausal, between the ages of 50–75 and followed for a minimum of 7 years to ensure cancer-free status; cancer-free status was based on an annual self-reported questionnaire.

Plasma samples from treatment-naïve cases and control subjects presenting with benign pelvic masses were drawn from the MD Anderson Gynecologic Cancer Bank. Cases were randomly selected and not prioritized based on CA125 values.

Initial testing was performed using plasmas from 41 early stage (I-II) cases, 75 late-stage (III-IV) cases, 71 healthy controls and 72 patients with benign pelvic masses. The validation set consisted of an independent set of plasmas from 61 early-stage cases and 71 healthy controls. Subject characteristics are provided in Table 1. Detailed information on the benign pelvic masses as well as histological subtype of ovarian cancers amongst cases is provided in Table A1 in the Appendix A.

### 2.2. Metabolomics Analysis

Detailed information is provided in the Appendix A. Semi-quantitative measurement of plasma polyamines was conducted on a Waters Acquity™ UPLC system with 2D column regeneration (I-class and H-class) coupled to a Xevo G2-XS quadrupole time-of-flight (qTOF) mass spectrometer. Chromatographic separation was performed using HILIC (Acquity™ UPLC BEH amide, 100 Å, 1.7 µm 2.1 × 100 mm) and C18 (Acquity™ UPLC HSS T3, 100 Å, 1.8 µm, 2.1 × 100 mm) columns at 45 °C. Mass spectrometry data were acquired in sensitivity, positive electrospray ionization mode. Acquisition was carried out with instrument auto-gain control to optimize sensitivity during sample acquisition; data processing was performed as previously described [10,11].

Four polyamines (diacetylspermine (DAS), acetylspermidine (AcSpmd), diacetylspermidine (DiAcspmd), and N-(3-acetamidopropyl)pyrrolidin-2-one (N3AP)) were detected and quantified in plasmas of cases and controls (Table A2 in the Appendix A). Coefficient of variation (CV) values for the measured polyamines in quality controls are provided in Table A3 in the Appendix A. On average, CV values for measured polyamines in quality control samples were below 13%.

### 2.3. Measurement of CA125 Levels

Automated immunoassay kits for determining the concertation of CA125 antigen were purchased from Roche Diagnostics USA (Indianapolis, IN, USA).

### 2.4. Statistical Analyses

For two-class comparisons, statistical significance was determined using the Wilcoxon rank sum test. Statistical significance was determined at *p*-values <0.05. Receiver operating characteristic curves were generated using R (R version 3.6.0). The 95% confidence intervals presented for the individual performance of each biomarker were based on the bootstrap procedure in which we re-sampled with replacement separately for the controls and the diseased 1000 bootstrap samples. To identify the optimal combination of markers amongst CA125 and four other polyamines, we implemented Lasso regression. [12] Ten-fold cross-validation on the training dataset with AUC as the measurement of classification performance was incorporated to find the best tuning parameters (λ). This parameter yielded CA125, DAS, and N3AP as the best combination of biomarkers; a biomarker panel using CA125, DAS, and N3AP were subsequently derived using a logistic regression model. The estimated AUC of the proposed metabolite panel was derived by using the empirical ROC estimator of the linear combination corresponding to the aforementioned model. 

Confusion matrices were utilized to describe the classification model of the 3-marker panel or CA125 alone at a 99% specificity cutoff. Rows of the matrix display the predicted classes (case or control) whereas columns represent the actual classes (case or control). 

To test whether the 3-marker panel yielded statistically significant classifier improvements over CA125, the McNemar exact test was applied to compare two binomial proportions for patients with two different biomarker scores [13]. Herein, a 2 × 2 table was generated wherein the first cell represents the number of patients that both markers predict correctly (a), the second one represents the number of patients correctly identified by CA125 and misclassified by the 3-marker panel (b), the third one represents the number of patients misclassified by CA125 but correctly identified by the 3-marker panel (c) and the last cell represents the number of patients misclassified by both markers (d). Therefore, the null and alternative hypothesize are as follows:H_0: P_b = P_c
H_a: P_b < P_c

Herein, P_i denotes the probability of occurrence in cell i. An exact binomial test was used to achieve *p*-value [14].

All statistical tests were two-sided unless specified otherwise.

## 3. Results

### 3.1. Polyamine Levels in Plasma of Ovarian Cancer Patients and Model Development

To determine whether a polyamine signature is associated with ovarian cancer, we screened polyamine levels in plasma from 116 OvCa patients (41 early stage + 75 late stage, 91 serous, 25 non-serous) and 71 healthy controls (Test Set) using ultrahigh performance liquid chromatography mass spectrometry (Table 1). We additionally evaluated the levels of polyamines in plasma from 72 patients presenting with diverse benign conditions to determine the specificity of polyamines for OvCa (Table 1; detailed information is provided Table A1).

A total of four polyamines (DAS, AcSpmd, DiAcspmd, and N3AP) were detected and quantified in plasmas of cases and controls (Table A2 in the Appendix A). AUCs of individual polyamines for distinguishing OvCa from healthy controls ranged from 0.74–0.88 (Figure 1A; Table A4 in the Appendix A). Measured polyamines were generally positively correlated to each other (Figure A1 in the Appendix A). Of the measured polyamines, DAS exhibited the highest classification performance for distinguishing ovarian cancer cases from healthy controls (AUC: 0.88 (95% C.I., 0.84–0.93)) or patients with benign pelvic masses (AUC: 0.78 (95% C.I., 0.72–0.85)) (Figure 1A,B; Table A4 in the Appendix A). AUCs of DAS for distinguishing all serous cases from healthy controls or subjects with benign pelvic masses were 0.90 and 0.80, respectively (Table A5 in the Appendix A). When considering only early-stage serous cases, DAS yielded AUCs of 0.81 and 0.70 in comparison to healthy controls or subjects with benign pelvic masses, respectively (Table A5 in the Appendix A). AUCs of DAS for distinguishing non-serous OvCa cases from healthy controls or subjects with benign pelvic masses or healthy controls were 0.83 and 0.71, respectively (Table A5 and Figure A2 in the Appendix A).

Using logistic regression models, we determined whether the combination of polyamines plus CA125 would yield improved classification performance in differentiating early-stage OvCa cases from controls (healthy subjects + subjects with benign pelvic masses) when compared to CA125 alone. The combination of plasma DAS + N3AP + CA125 was identified as the best panel with an AUC point estimate of 0.84 (95% C.I., 0.77–0.92) (Figure 2A). Given the low prevalence of ovarian cancer (11.4 in every 100,000 people), a screening test must achieve high sensitivity at high specificity to avoid unacceptable levels of false-positive results. Therefore, we evaluated the sensitivity of the 3-marker panel (DAS + N3AP + CA125) in comparison to CA125 alone at corresponding specificities of 99%, 98.5% and 97% (Table A6 in the Appendix A). Compared to CA125 alone, the 3-marker panel yielded statistically significantly improved sensitivity at 99% and 98.5% specificity (1-sided McNemar Exact test P: 0.006 and 0.04, respectively) (Table A6 in the Appendix A). A confusion matrix describing the performance of the classification model corresponding to the 3-marker panel and CA125 alone at 99% specificity illustrates that the 3-marker panel correctly identified 19 out of 44 early-stage OvCa cases (46.3% sensitivity) whereas CA125 alone correctly identified 10 out of early-stage 41 OvCa cases (24.3% sensitivity) (1-sided McNemar exact test P: 0.006) (Figure 2B,C). 

In our study, correlation analyses between DAS, N3AP, and CA125 levels (which comprise the 3-marker panel) with ages amongst healthy controls were non-statistically significant (data not shown). Age-adjusted ROC analyses revealed that the 3-marker panel of DAS + N3AP + CA125 (age-adjusted AUC: 0.80 (95% CI: 0.71–0.88)) yielded improved classifier performance as compared to CA125 alone (age-adjusted AUC: 0.70 (95% CI: 0.58–0.82), Delong’s comparison of AUCs 1-sided P: 0.002). These findings imply that age is not a confounder in the context of the current study.

### 3.2. Model Validation in an Independent Cohort of Early Stage Ovarian Cancer Patients

Further validation of the polyamine metabolites individually and the fixed 3-marker panel consisting of DAS + N3AP + CA125 was performed in an independent set of plasma samples consisting of 61 early-stage OvCa cases (serous = 28, non-serous = 33) and 71 healthy controls (Validation Set; Table 1). AUCs for individual polyamine metabolites ranged from 0.57–0.84 (Figure 3). Plasma DAS exhibited the highest AUC point estimate for delineating OvCa cases from controls (AUC: 0.84 (95% C.I., 0.77–0.91)) (Figure 3). Classification performance of plasma DAS in distinguishing serous (*n* = 28) and non-serous cases (*n* = 33) from healthy controls was 0.84 (95% C.I., 0.75–0.93) and 0.84 (95% C.I., 0.75–0.92), respectively (Table A7 and Figure A3 in the Appendix A).

In the validation set, the fixed 3-marker panel yielded an AUC of 0.95 (95% C.I., 0.92–0.99) whereas CA125 yielded an AUC of 0.96 (95% C.I., 0.94–0.99). Sensitivity of the 3-marker panel using fixed beta-coefficients derived from the test set at corresponding specificities of >99%, 98.5%, and 97% for the validation set were 73.7, 78.6, and 83.6, respectively (Table A6 in the Appendix A). Compared to CA125 alone, the 3-marker panel yielded statistically significantly improved sensitivity at >99% specificity (3-marker panel sensitivity: 73.7%, CA125 sensitivity: 62.2%, 1-sided McNemar exact test P: 0.02) (Table A6 in the Appendix A). A confusion matrix describing the performance of the classification model corresponding to the 3-marker panel and CA125 alone at >99% specificity shows that the 3-marker panel correctly identified 45 out of 61 early-stage OvCa cases in the validation set whereas CA125 alone correctly identified 38 out of 61 early-stage OvCa cases, which corresponds to capturing 30.4% of cases missed by CA125 alone (Figure 4A,B). 

A CA125 cutoff value of 35 U/mL is considered the upper limit of “normal” [15]. In the validation set, a cutoff value of 35 U/mL captured 53 out of 61 cases (86.9% sensitivity) with 2 false-positives (97.2% specificity). If considering cases “negative” for CA125 (defined as ≤ 35 U/mL), the fixed 3-marker panel identifies an additional 2 of the 8 early-stage cases (25.0% sensitivity) without any additional false-positives (Figure A4 in the Appendix A). Notably, these two additional cases identified by the 3-marker panel were stage I high-grade serous carcinomas.

## 4. Discussion

We demonstrated and validated that polyamines are statistically significantly elevated in plasmas of ovarian cancer cases in comparison to controls using two independent case-control cohorts. Of the measured polyamines, DAS exhibited the highest AUC for distinguishing cases from healthy controls or subjects presenting with benign pelvic masses. Importantly, we demonstrated that a 3-marker panel consisting of DAS + N3AP + CA125 yielded improved sensitivity at >99% specificity in comparison to CA125 only, resulting in capturing early-stage cases that were missed by CA125 alone in the validation set. Classifier performance of polyamines was higher in late-stage compared to early-stage cases, implicating that the polyamine levels reflect disease burden. Further, polyamines were elevated in both serous and non-serous cases. Recent reports indicated marked heterogeneity in clinical outcomes of ovarian cancer histological subtypes with non-serous subtypes exhibiting survival rates similar to or worse than that of high-grade serous carcinomas [16]. A blood test that can broadly identify ovarian cancers is, therefore, desirable.

There are potential limitations inherent to the available cohorts used in the study. Healthy controls tended to be older than cases. However, the 3-marker panel retained improved classifier performance compared to CA125 alone when adjusting AUCs for age, indicating that age is not a confounder variable in our analyses. Our study has a moderate sample size of early-stage OvCa cases. We emphasize that polyamines, DAS, in particular, yielded a good classifier performance for distinguishing early-stage OvCa cases from healthy controls in both the test and validation set. The performance of polyamine markers was reduced when considering subjects with benign pelvic masses suggestive that the presence of benign conditions may result in potential false-positives. Nevertheless, our findings demonstrated that polyamines, particularly DAS, were capable of distinguishing early-stage OvCa from subjects presenting with benign pelvic masses. 

Integrated genome analysis has reported that the *MYC* gene is amplified in 30–40% of human ovarian tumors [8,17,18,19]. In addition to *MYC*, copy-number amplification and/or overexpression of *MYCL1* and *MYCN* have also been reported in ovarian tumors [20,21,22]. We and others have demonstrated that the polyamine metabolizing enzymes ODC1, SRM and SMS are transcriptionally regulated by MYC [10,23,24,25] thereby providing a link between our polyamine signature and regulation via oncogenic MYC. In the context of our study, information regarding MYC status in tumors of subjects for which plasma polyamine levels were analyzed was not available thus precluding correlative analyses.

Prior studies have examined polyamine levels in urine and in one study, urine polyamines were found to be elevated among subjects with ovarian cancer. Similar to our findings, DAS exhibited the best classification performance for delineating ovarian cancer cases from individuals with benign disease and was found to be associated with disease progression [26].

In conclusion, we have identified an MYC-driven polyamine signature that reflects the early pathogenesis of ovarian cancer. Given the substantial interest in developing strategies for cancer detection and given the limited performance of CA125, our validation of a polyamine signature for early-stage ovarian cancer that complements CA125 provides supportive evidence for the utility of aberrant polyamine metabolism for cancer detection. Our findings provide a basis for the inclusion of the 3-marker panel on validation studies that encompass other biomarker candidates [26,27,28].

## Figures and Tables

**Figure 1 cancers-13-00913-f001:**
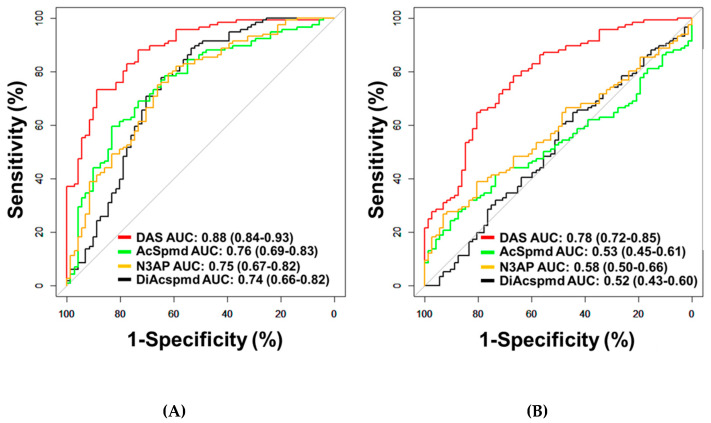
Classification performances of plasma polyamines in the Test Set. (**A,B**) Area under the curve (AUC) of DAS for delineating all cases (*n* = 116) from healthy controls (*n* = 71) (**A**) or patients with benign pelvic masses (*n* = 72) (**B**).

**Figure 2 cancers-13-00913-f002:**
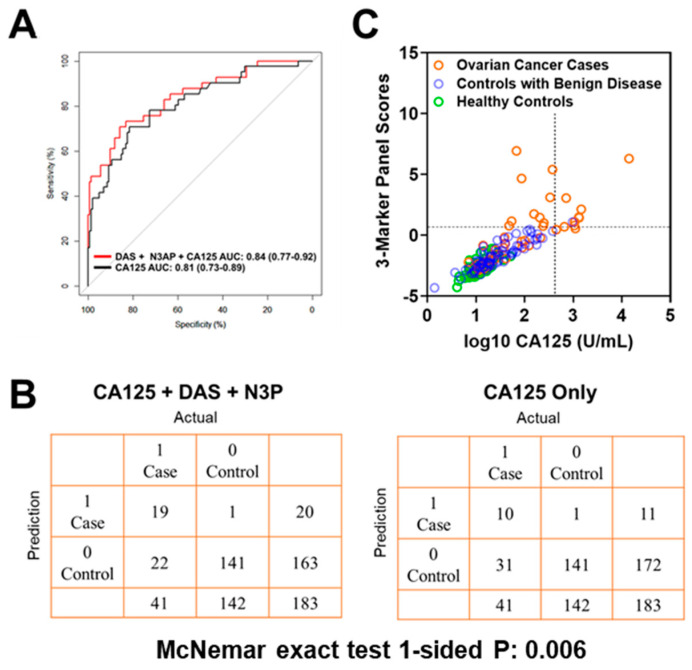
Classification performance of the 3-marker panel and CA125 in the Test Set. (**A**) Area under the curve for the 3-marker panel consisting of DAS + N3AP + CA125, and CA125 only. (**B**) Confusion matrix describing the performance of the classification model corresponding to the 3-marker panel and CA125 alone at 99% specificity. Statistical significance was determined by 1-sided McNemar exact test. (**C**) Scatter plot illustrating the distribution of the 3-marker panel scores (Y-axis) and log10 CA125 values (X-axis). Broken lines represent 99% specificity cutoffs.

**Figure 3 cancers-13-00913-f003:**
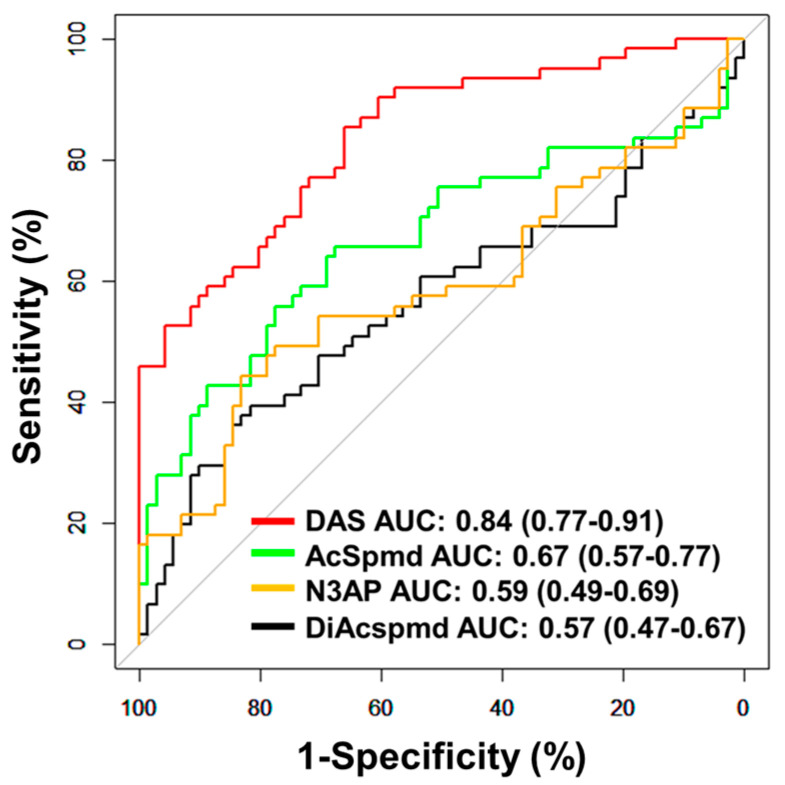
Classification performances of plasma polyamines in the validation set. The area under the curve (AUC) of individual polyamines for delineating all cases (*n* = 61) from healthy controls (*n* = 71).

**Figure 4 cancers-13-00913-f004:**
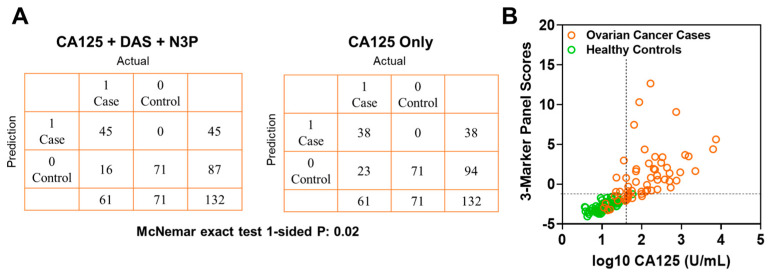
Classification performance of the 3-marker panel and CA125 in the validation set. (**A**) Confusion matrix describing the performance of the classification model corresponding to the 3-marker panel and CA125 alone at 99% specificity. Statistical significance was determined by 1-sided McNemar exact test. (**B**) Scatter plot illustrating the distribution of the 3-marker panel scores (Y-axis) and log10 CA125 values (X-axis). Broken lines represent >99% specificity cutoffs.

**Table 1 cancers-13-00913-t001:** Patient characteristics for the Test Set and Validation Set.

Patient Characteristics for Specimen Sets	Test Set			Validation Set	
	Cases	Controls #1 †	Controls #2 ‡	Cases	Controls #1 †
Number of Subjects	116	71	72	61	71
Age (mean +/− stdev)	58 +/− 13	69 +/− 7	56 +/− 13	57 +/− 15	65 +/− 9
CA125 (u/mL), median (25th/75th percentile)	224 (87/561)	12 (9/16)	23 (13/59)	102 (45/321)	11 (8/17)
Serous					
Stage I, N (%)	11 (9.5)	-	-	13 (21.3)	-
Stage II, N (%)	5 (4.3)	-	-	15 (24.6)	-
Stage III, N (%)	64 (55.2)	-	-	-	-
Stage IV, N (%)	11 (9.5)	-	-	-	-
Non-Serous					
Endometrioid					
Stage I, N (%)	10 (8.6)	-	-	16 (26.2)	-
Stage II, N (%)	2 (1.7)	-	-	6 (9.8)	-
Mucinous					
Stage I, N (%)	3 (2.6)	-	-	6 (9.8)	-
Stage II, N (%)	1 (0.9)	-	-		
Clear Cell Carcinoma					
Stage I, N (%)	6 (5.2)	-	-	3 (4.9)	-
Stage II, N (%)	2 (1.7)	-	-	1 (1.6)	-
Other					
Transitional cell carcinoma (Stage I)	1 (0.9)	-	-	1 (1.6)	-

† Controls #1- Healthy Controls. ‡ Controls #2- Subjects with benign pelvic masses.

## Data Availability

Relevant data supporting the findings of this study are available within the Article and Appendix A or are available from the authors upon reasonable request.

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
