# Peer review of "A MYC-Driven Plasma Polyamine Signature for Early Detection of Ovarian Cancer"

_cancers, 2021, doi:10.3390/cancers13040913_

Round 1
Reviewer 1 Report
The authors used ultra-high performance liquid chromatography mass spectrometry to quantify four polyamines in plasma and identify a polyamine signature that was able to complement CA125 in identifying more ovarian cancer cases that would have been missed by CA125 alone. A 3-marker panel significantly improved sensitivity versus CA125 only. Despite the relatively small number of patients examined, the results were highly significant and further validated in an independent plasma set. The study addresses a need in the field to identify additional markers to detect early stage ovarian cancer that would complement the performance of CA125. By identifying a plasma polyamine signature associated with the disease, the study is exciting, important and the signature warrants further investigation in ovarian cancer. The results are presented in a clear fashion; the paper is well written.
Minor
MYC has widespread amplification in ovarian cancer, and its amplification typically correlates with high-level expression of the MYC oncogene. In the corresponding tumors from the patients examined in this study, do the authors have information on MYC alterations in any of the patients examined? If so, it would be interesting to include this information.
Including a paragraph on the frequency of altered MYC in ovarian cancer in the Discussion would be of interest to the readership of Cancers.
Author Response
Minor
MYC has widespread amplification in ovarian cancer, and its amplification typically correlates with high-level expression of the MYC oncogene. In the corresponding tumors from the patients examined in this study, do the authors have information on MYC alterations in any of the patients examined? If so, it would be interesting to include this information.
Response: Information regarding MYC status in tumors of subjects for which we have analyzed plasma polyamine levels was not available thus precluding such correlative analyses. We have now commented upon this limitation in the discussion section.
Including a paragraph on the frequency of altered MYC in ovarian cancer in the Discussion would be of interest to the readership of Cancers.
Response: We have now included additional content regarding frequency of MYC alterations in ovarian cancer tumors in the discussion section of the revised manuscript.
Reviewer 2 Report
In this paper, Dr. Fahrmann et al. have demonstrated the expecting possibility that plasma polyamine signature would be used as additional markers for detecting and distinguish early stage ovarian cancer with combined use of CA125, which is known as a typical marker of ovarian cancer. Impressively and importantly, the authors demonstrated in Figure 4 that 3-marker panel consisting of DAS + N3AP + CA125 yielded markedly improved sensitivity, compared with single use of CA125 marker.
Their idea of use of polyamines, research design, and interpretation against research data are very logical and reasonable. The results are very important and suggestive.
I found several confusion against Figure and Table numbers. Distinguish Figures and Tables in Text (Figure 1, 2,,,,) and Appendix (Figure A1, A2,,,,,).
Author Response
I found several confusion against Figure and Table numbers. Distinguish Figures and Tables in Text (Figure 1, 2,,,,) and Appendix (Figure A1, A2,,,,,).
Response: To avoid confusion, we have now included an additional statement that supplementary figures and tables are provided “in the Supplementary Appendix”.
Reviewer 3 Report
In their manuscript entitled “A MYC-driven plasma polyamine signature for early detection of ovarian cancer”, Fahrmann and co-workers investigate a biomarker approach based on polyamine signature in blood to detect ovarian cancers at an early stage.
Starting from the fact that the most widely used tumour marker CA125 is not suitable for detection of early cancer, the authors emphasise the need for additional markers and propose polyamines based on previous observations.
Being MYC an oncogenic driver of ovarian cancer, the authors start from their finding that MYC regulates polyamine metabolism in triple-negative breast cancer (TNBC). Moreover, a plasma polyamine signature was associated with TNBC development and progression. Given that TNBC and ovarian cancer share genetic features, the authors hypothesised that this approach might be suitable for ovarian cancer as well. Using mass spectrometry, four polyamines were quantified in plasma from patients in a test set (116 cases and 143 controls including healthy controls and subjects with benign pelvic masses). The encouraging findings were validated in an independent set of subjects evaluating 61 early-stage ovarian cancer cases and 71 healthy controls (Test Set). A polyamine signature consisting of diacetyl-32 spermine + N-(3-acetamidopropyl)pyrrolidin-2-one in combination with CA125 improved the marker sensitivity to >99% specificity when compared with CA125 alone and identified 30.4% of cases that would have been missed by using CA125 alone. The authors concluded that a MYC-driven plasma polyamine signature was associated with early-stage ovarian cancer thus complementing CA125 in its diagnostic value.
This is very sound clinical research starting from the observation that oncogenic MYC drives an aberrant polyamine signature, which was validated with considerable analytical effort towards clinical application. Although more data are needed and will certainly be collected, this manuscript provides data about the power of combined biomarker analysis expected to increase in several cancer entities. The inclusion of benign entities is a particularly useful feature of this investigation.
The referee has no major objections to the current work.
Minor issues
- The authors describe four polyamines; is this a selection of more polyamines detected? If so, how was the selection performed?
- How stable are the analytes in plasma considering that clinical samples have highly different processing times in daily routine?
- Have the authors ever looked whether the polyamines are also present in extracellular vesicles, where compound may be selectively enriched? If not, simply disregard this comment.
Author Response
Minor issues
- The authors describe four polyamines; is this a selection of more polyamines detected? If so, how was the selection performed?
Response: Based on our metabolomics analyses, we were able to detect and quantify four polyamines: diacetylspermine (DAS); acetylspermidine (AcSpmd); diacetylspermidine (DiAcspmd); and N-(3-acetamidopropyl)pyrrolidin-2-one (N3AP) in plasmas of cases and controls (Table A2 in the Supplementary Appendix). Other polyamines, such as putrescine, spermidine, and spermine, were not detected.
- How stable are the analytes in plasma considering that clinical samples have highly different processing times in daily routine?
Response: We understand the Reviewer’s concern. We note that all biospecimen were processed at a central site following a standardized protocol; EDTA-plasmas were stored in -80°C until use. Metabolomics analyses were performed using established standard operating procedures and quality controls included to ensure data reproducibility. We note that we have previously undertaken studies to characterize the stability of profiled metabolites of interest and determined that they are not significantly impacted by freeze/thaw cycles or processing time latency (up to 8 hrs).
Briefly, for these sample stability trials, blood samples (n=3) were processed immediately or after standing at room temperature for 4 or 8 hours prior to being processed. Post processing, EDTA plasma either immediately underwent metabolite extraction as per our standard operating procedures, or was subjected to 1 or 3 freeze-thaw cycles prior to metabolite extraction. All samples were run in a single batch and untargeted metabolomics profiling conducted as described in the manuscript.
- Have the authors ever looked whether the polyamines are also present in extracellular vesicles, where compound may be selectively enriched? If not, simply disregard this comment.
Response: We have not assessed whether polyamines are present in extracellular vesicles.